# The Contest of Nanoparticles: Searching for the Most Effective Topical Delivery of Corticosteroids

**DOI:** 10.3390/pharmaceutics15020513

**Published:** 2023-02-03

**Authors:** Aneta Kalvodová, Kristýna Dvořáková, Eliška Petrová, Bozena B. Michniak-Kohn, Jarmila Zbytovská

**Affiliations:** 1Department of Organic Technology, Faculty of Chemical Technology, University of Chemistry and Technology Prague, Technická 5, 166 28 Prague, Czech Republic; 2Center for Dermal Research (CDR), Life Sciences Building, Rutgers University, Piscataway, NJ 08854, USA; 3Department of Pharmaceutical Technology, Faculty of Pharmacy, Charles University, Heyrovského 1203, 500 05 Hradec Králové, Czech Republic

**Keywords:** lipid nanocapsules, PLGA nanoparticles, ethosomes, hydrocortisone, hydrocortisone-17-butyrate, dermal and transdermal delivery

## Abstract

Owing to their complicated pathophysiology, the treatment of skin diseases necessitates a complex approach. Conventional treatment using topical corticosteroids often results in low effectiveness and the incidence of local or even systemic side effects. Nanoformulation of potent anti-inflammatory drugs has been selected as an optimal strategy for enhanced topical delivery of corticosteroids. In order to assess the efficiency of various nanoformulations, we formulated hydrocortisone (HC) and hydrocortisone-17-butyrate (HCB) into three different systems: lipid nanocapsules (LNC), polymeric nanoparticles (PNP), and ethosomes (ETZ). The systems were characterized using dynamic light scattering for their particle size and uniformity and the morphology of nanoparticles was observed by transmission electron microscopy. The nanosystems were tested using ex vivo full thickness porcine and human skin for the delivery of HC and HCB. The skin penetration was observed by confocal microscopy of fluorescently labelled nanosystems. ETZ were proposed as the most effective delivery system for both transdermal and dermal drug targeting but were also found to have a profound effect on the skin barrier with limited restoration. LNC and PNP were found to have significant effects in the dermal delivery of the actives with only minimal transdermal penetration, especially in case of HCB administration.

## 1. Introduction

Atopic dermatitis (AD) or atopic eczema is one of the most prevalent types of skin dermatoses. AD affects predominantly children (the average prevalence in population is reported to be between 10–20% [1]); however, a certain percentage of cases prevail into later years and the evidence suggests that its pervasiveness continues to increase in the population [2]. AD is a chronically relapsing dry skin condition characterized by persistent itchy rashes, dry skin or lichenification, and skin thickening with hyperkeratosis [3]. The impaired skin barrier is often inhabited by bacterial, viral, or fungal infections, leading to inflammatory episodes, further weakening the normal skin structure [4]. The disease impairs the quality of patients’ lives, worsens the sleep time, and also presents a significant economic burden [5].

Due to the variety of AD triggers, a complex methodology for treatment is required [6]. It should include not only the protection of the skin and restoration of the skin barrier, but should also target the inflammatory processes in the deeper layers of the skin. The conventional treatment for mild AD cases includes the use of ointments, creams, and other emollients [7]. These hydrating formulas aim at improving the stratum corneum (SC) lipid film and the skin barrier function. Another approach to impaired skin restoration is the application of suitably formulated skin lipids. Our previous work regarding the development and application of cerosomes proved the ability of liposomally formulated ceramides and other skin lipids to restore chemically induced dermatitic skin barriers [8,9]. These approaches, however, do not target the auto-immune processes promoting the disease. For these more severe cases, the pharmacological therapy is needed and for a long time, the most frequently prescribed drugs have been topical corticosteroids (TCs) [10]. TCs can be classified into several groups by their potency and capability to cause vasoconstriction [11]. This enables broad possibilities for the dosages and administration of TCs. However, caution is needed when using highly potent TCs on sensitive skin areas, where there is an increased probability of local or systemic side effects, such as high blood pressure, glaucoma, skin thinning, or muscle weakness [12,13,14,15].

A typical anti-inflammatory active is hydrocortisone (HC). The only naturally occurring and directly active corticosteroid. It is also commonly used as a model drug representing the ones needing localized topical targeting with minimal systemic absorption [16]. Two of the most important pharmacological actions of HC are the anti-inflammatory and immunosuppressive effects causing the inhibition of prostaglandin and leukotriene synthesis, crucial in the treatment of AD or psoriasis [17]. HC being one of the first of corticosteroids used is now replaced by derivatives with higher potency or more suitable physico-chemical properties [18], such as hydrocortisone-17-butyrate (HCB). HCB is a more lipophilic compound with a longer elimination half-time in the body. It is widely used in prescription ointments for dermatological problems.

The topical application of actives, such as HC and HCB, is often faced with challenges which are typical of larger water-insoluble molecule permeation across the skin barrier. The dermal bioavailability of such drugs delivered by a topical formulation can be as low as 5–10% [19]. The delivery can be enhanced by the incorporation of the actives into carriers that will promote the passage through the skin barrier [20,21]. There have been several formulations developed for HC and HCB topical delivery [22,23,24,25,26]. There is a variety of nanoformulations for corticosteroids, but the data obtained varies depending on experimental conditions, resulting in non-uniformity.

For the purpose of better understanding the corticoid dermal bioavailability, we developed and optimized three distinct nanoparticulate systems containing HC or HCB. Lipid nanocapsules (LNCs) were selected for their biocompatible composition, ensuring the improved dispersibility and enhanced entrapment of predominantly lipid soluble drugs. LNC are able to efficiently promote the skin (trans)dermal penetration, increase the skin hydration by their occlusive effect, and ensure sustained drug release. Currently, LNC are being used as carriers for anti-dermatitic [27] and anti-cancer treatments [28]. The next nanovectors in question were polymeric nanoparticles (PNP). PNP are made of natural synthetic polymers with a great possibility for particle size selection. They are capable of high drug loading, consecutively leading to an intensive transcutaneous drug flux. Despite their issues with short colloidal stability, PNP are widely studied as carriers for anti-inflammatory [29], anti-fungal [30], or immunosuppressive [31] actives. The last studied nanovesicles were ethosomes (ETZ) because of their extensive positive effects on the drug release to various layers of the skin. The high ethanol content that could raise questions about skin barrier interaction is however balanced by their reported effectiveness in topical delivery of anti-inflammatory [32] or anti-cancer moieties [33]. The overview of advantages and disadvantages of presented nanosystems is stated in Table 1.

The developed nanoparticulate systems were characterized for their particle size, morphology, drug content, and stability and their efficacy was compared ex vivo on porcine and human skin. To further characterize the mode of action of the nanoparticles, we evaluated the effect that different formulations have on the skin lipid arrangement and trans-epidermal water loss (TEWL). The permeation depth of the formulation components was visualized by confocal laser scanning microscopy.

## 2. Materials and Methods

### 2.1. The Materials

Phospholipon 90G was obtained from Lipoid GmbH (Ludwigshafen, Germany). Kolliphor HS 15, isopropyl myristate (IPM), Resomer RG 503 (poly-D,L-lactic-co-glycolic acid, PLGA), polyvinyl alcohol (PVA), hydrocortisone (HC), hydrocortisone-17-butyrate (HCB), phosphate buffer saline (PBS) tablets, gentamicin sulphate, NaCl, KBr, fluorescein sodium, and DMAP catalyst were purchased from Merck KGaA (Darmstadt, Germany). The 18:1–12:0 NBD-PC was obtained from Avanti polar lipids (Birmingham, UK). Acetonitrile, methanol, and ethanol were obtained from Penta s.r.o. (Prague, Czech Republic). Acetic acid was purchased from Lach-Ner s.r.o. (Neratovice, Czech Republic). Milli-Q water was distilled at the Department of Organic Technology, UCT Prague.

### 2.2. Nanosystem Formulation Preparation

#### 2.2.1. Lipid Nanocapsules

Lipid nanoparticles (LNCs) containing corticoids were prepared using the phase inversion temperature (PIT) method [34] with modifications needed to formulate stable carriers for HC and HCB. The composition of the final formulation is presented in Table 2. Initially, the weighed active pharmaceutical ingredient (API) was dissolved in the needed amount of a IPM, Phospholipon 90G (PL), and Kolliphor HS15 mixture and heated on a magnetic stirrer to 85 °C for at least 10 min to ensure the dissolution of the drug and homogenization of the excipients. The mixture was allowed to cool to ambient temperature and then NaCl and the first portion of PBS were added. The second part of the PBS was put into freezer for later use. The primary emulsion was equipped with a thermometer and placed back onto the magnetic stirrer. Three heating cycles from 50 to 85 °C were applied to ensure a complete homogenization of the emulsion. After the last heating to 85 °C, the mixture was let to cool down just few degrees above the PIT of the surfactants (around 75 °C) and the cold PBS portion from the freezer was immediately added. This cold shock induced the creation of nanoparticles and the encapsulation of a drug into them [35]. The LNCs were then stirred until they reached laboratory temperature. The final product containing 0.1% API was kept in darkness at 25 °C.

#### 2.2.2. Polymeric Nanoparticles

Polymeric nanoparticles (PNPs) were prepared by the nanoprecipitation method according to [36]. At first, organic and aqueous phases were prepared separately. The organic phase consisted of the drug (HC: 9 mg; HCB: 3 mg) and poly-D,L-lactic-*co*-glycolic acid (PLGA) polymer (HC: 30 mg; HCB: 45 mg) dissolved in 4 mL of acetone. The aqueous phase was a solution of polyvinyl alcohol (PVA; HC: 0.5%; HCB: 2%) in distilled water (*w*/*v*). The complete composition of both PNP systems is stated in Table 3. For the preparation of nanoparticles, 20 mL of the aqueous phase was placed in a beaker, equipped with stirrer, and mixed at 500 rpm at a room temperature on a mixer. The organic phase was aspirated into a glass high-pressure syringe and by a linear pump (New Era Pump Systems, Farmingdale, NY, USA), continuously added in a rate of 0.5 mL/min into the aqueous phase. Then, the acetone was evaporated from the system on a rotary evaporator (BÜCHI Labortechnik AG, Flawil, Switzerland) at 40 °C, 80 rpm. This exchange in the solution composition ensured the precipitation of the polymers at the phase interface and subsequently the creation of PLGA nanoparticles. Finally, the formulation was concentrated to the desired concentration 0.1% for HC and 0.05% for HCB (*w*/*v*) using the rotary evaporator. The PNPs were stored at 4 °C in the dark.

#### 2.2.3. Ethosomes Preparation

Ethosomes were prepared by the classic cold method [37,38]. Phospholipon 90G (3% *w*/*v* in total formulation) was dissolved along with HC or HCB in ethanol. The mixture was maintained under magnetic stirring at 700 rpm. An appropriate amount of PBS needed to obtain a 1:1 (*v*/*v*) mixture with ethanol in the final formulation was added linearly by using a microinjection linear pump at the rate of 200 µL/min. The resulting mixture was stirred at 700 rpm for 5 min to obtain the ethosomal colloidal suspensions. The composition of the best ethosomal formulation, including PL and dispersion medium, is reported in Table 3. The final formulations containing 0.1% HC or HCB were stored at 25 °C.

The composition of our presented nanosystems was selected to yield consistent reproducible particles. The criterion for successful repeatable formulation was set to be a less than 10% difference in the mean particle size and PDI [39]. All of the compositions stated in Table 1 and Table 2 passed this threshold.

#### 2.2.4. Fluorescein-Labelled Nanoparticles

For the purposes of the detection of PLGA nanoparticles in the skin after the application, the polymer was labelled by a fluorescent probe. The process was carried out according to [40]. Briefly, the polymer (3 g), together with 0.05 g fluorescein sodium (FNa) and the catalyst (0.04 g DMAP), was weighed into a round bottom flask and dissolved in 30 mL acetonitrile. The mixture was left on a mixer at 1000 rpm for 24 h at laboratory temperature. The polymer was then precipitated by the addition of excess distilled water, separated from the aqueous phase, and repeatedly washed with dichloromethane. The organic solvent was evaporated on a rotary evaporator to yield the resulting fluorescein-labelled PLGA. The labelled nanoparticles were then prepared by the same method as described above (see Section 2.2.2).

Fluorescently labelled ETZ and LNC were prepared using the partial replacement of Phospholipon^®^ 90G (0.05% *w*/*w*) by the labelled lecithin (NBD-PC) to their composition.

### 2.3. Nanoparticulate Systems Characterization

#### 2.3.1. Colloidal Properties

Initially, all samples were evaluated by visual observation. Signs of aggregation, API crystallization, non-dissolved lipids, or other flaws were noted and any sample showing the aforementioned was eliminated from further testing. The same parameters were observed using optical microscopy (Olympus CX 43, Tokyo, Japan). To determine the size and zeta potential (ZP), samples were analyzed by dynamic light scattering (DLS) on Zetasizer Nano ZS (Malvern instruments, Worcestershire, UK). For the measurements, 10 µL of the sample were dispersed in 1 mL of double-distilled water. The scattering angle was set to 173°, temperature was 25 °C, and the analysis consisted of three measurements with five scans. The particle size was measured as the Z-average value, which was defined as the intensity weighed mean hydrodynamic diameter. The size dispersity was determined as the polydispersity index (PDI). ZP was measured in a disposable folded capillary cell (DTS 1070). Henry’s function approximation and Smoluchowski model were used for the approximation. The number of replicates was three and the measurement duration was set automatically, but a minimal run number was 12 and a maximal was 100. To check the stability of the samples, all the characteristics were measured repeatedly in defined time-periods until a significant rise in either size or PDI was observed.

#### 2.3.2. Encapsulation Efficiency and Drug Load

Encapsulation efficiency (EE) and drug load (DL) of prepared nanoparticulate systems were obtained by the following procedure: 2 mL of the 10× diluted formulation was put into a filtration unit (Centrisart 1, MWCO 20,000, Sartorius, UK) and centrifuged for 4 h, 2000× *g*, 4 °C for LNCs and for 2 h, 2000× *g*, 10 °C for PNPs and ETZs. After the elapsed time, the supernatant was withdrawn and analysed by the HPLC-UV/VIS method (see Section 2.6). The EE and DL values were then calculated by Formulas (1) and (2), respectively [41]:(1)$$\mathrm{EE}\;\left(\%\right)=\frac{{w\;}_{\left(\mathrm{initially}\;\mathrm{added}\;\mathrm{API}\right)}-{\;w}_{\left(\mathrm{unencapsulated}\;\mathrm{API}\right)}\;}{w_{\left(\mathrm{initially}\;\mathrm{added}\;\mathrm{API}\right)}}\times 100$$
(2)$$\mathrm{DL}\;\left(\%\right)=\frac{{w\;}_{\left(\mathrm{entrapped}\;\mathrm{API}\right)}}{w_{\left(\mathrm{carriers}\right)}}\times 100$$
where w_(initially added API)_ is the total mass of the API weighed in the formulation, w_(unencapsulated API)_ is the mass of the API measured in the supernatant after the centrifugation, w_(entrapped API)_ is the amount of the API encapsulated in the system, and w_(carriers)_ is the weight of the excipients used in the particular formulation for the formation of carriers.

#### 2.3.3. Fourier-Transform Infra-Red Spectroscopy

To better estimate the interaction of HC or HCB with our nanosystems, the formulations were studied by attenuated total reflectance Fourier transform IR spectroscopy (FTIR), according to [8,42]. For this measurement, nanoparticles were freeze-dried and the lyophilizate was then tabletted with KBr on a table-top tablet press (Specac Ltd., Orpington, United Kingdom) by a 1-ton pressure. Tablets containing physical mixtures of HC or HCB with formulation components were prepared by the same process as controls. Collectively, these samples were prepared for FTIR: for lipid nanocapsules: LNC, HC-LNC, HCB-LNC, HC + IPM, HCB + IPM; for polymeric nanoparticles: PNP, HC-PNP, HCB-PNP, HC + PLGA, HCB + PLGA; and for ethosomes: ETZ, HC-ETZ, HCB-ETZ, HC + PL, HCB + PL. The resulting tablets were then placed on a single reflection MIRacle ZnSe crystal (PIKE technologies, Madison, WI, USA) equipped in an IR spectrometer (Nicolet iZ10, Thermo Scientific, Waltham, MA, USA). The spectra were generated by the co-addition of 64 scans collected at a resolution of 2 cm^−1^ and evaluated by OMNIC™ software (Thermo Scientific, Waltham, MA, USA). The spectra were normalized in the OriginPro software (OriginLab, Wellesley Hills, MA, USA) by its inbuilt “normalize” function in order to minimize the influence of variations in total intensity of the measurements.

#### 2.3.4. Transmission Electron Microscopy

In order to study the morphology and size of the nanosystems, the samples were studied by transmission electron microscopy (TEM) on a Jeol-JEM 1010 microscope (Jeol Ltd., Tokyo, Japan). For the analysis, the sample was diluted 1:100 with distilled water and 10 μL of the sample was transferred onto a carbon-formvar coated copper mesh grid. Water was then let to evaporate for approximately 20 min. The dried-out mesh grids were then stained for 6 min using 1% uranyl acetate solution. Samples were then measured with an accelerating voltage of 80 kV.

### 2.4. Ex Vivo Permeation Study

#### 2.4.1. Skin Sample Preparation

Two types of skin barrier were used for the permeation experiments. The first type was the full thickness porcine skin. The skin sheets were obtained by a blunt dissection of shaved dorsal side of a porcine earlobe. After the isolation, the skin was washed in PBS and stored in freezer at −18 °C for up to 3 months. Before use, the skin was thawed and cut into approximately 1.5 × 1.5 cm squares. The second type of skin barrier was full thickness human skin. The skin was obtained from Caucasian female individuals who underwent abdominal plastic surgery with the approval of the Ethics Committee of the Sanus Surgical Centre in Hradec Králové (No. 5/4/2018), according to the principles of the Declaration of Helsinki. Written informed consent was obtained from all individuals.

#### 2.4.2. Donor Sample Preparation

LNCs, PNPs, or ETZs carrying HC or HCB were prepared as described in Section 2.3, respectively. A 1% suspension (an infinite dose) of the corticosteroid in PBS was prepared as a control formulation. Prior to the permeation test, all formulations were kept incubated to 32 °C for 24 h to ensure a minimal effect of temperature gradient on the permeation rate.

#### 2.4.3. Permeation Experiment

To study the effect of the nanoformulation systems, Franz-type diffusion cells with an available diffusion area of 1 cm^2^ were used. The square fragments of thawed skin were mounted between the acceptor and donor compartment dermal side down and secured with a clamp. The acceptor part was filled with PBS (pH = 7.4) containing gentamicin sulphate (50 mg/L) in order to prevent microbial contamination of the samples. The Franz diffusion cells with mounted skin samples were placed in a water bath with a constant temperature of 32 °C and equipped with a magnetic stirrer to stir acceptor phase throughout the experiment. After the hydration equilibrium period of 1 h, the skin integrity was verified by initial TEWL (AquaFluxTM, Model AF200, London, UK) and impedance (Multimeter LCR 4080, Voltcraft, Germany) measurements of the intact skin. Then, 300 μL of the donor samples, shaken on a vortex immediately prior to use, were applied to the skin surface. The acceptor compartment and sampling ports were covered by Parafilm^®^ to minimize evaporation. The permeation experiments run for 48 h total. Samples of the acceptor phase were withdrawn at predetermined time intervals and replaced with the same volume of fresh buffer solution. The concentrations of HC and HCB were then determined by HPLC.

#### 2.4.4. Entrapment of API in the Skin

To establish the level of drug concentrated in the skin after the topical application, the drug was extracted from the skin as follows: the permeated areas were cut out of the skin and washed thoroughly with PBS. The skin samples were then tape-stripped two times by a consistent hand pressure to ensure the removal of the donor sample left in the crevices of the skin surface. The samples were weighed and extracted (300 rpm, 24 h) into the respective HPLC mobile phase for each API. The extraction medium was then sampled into HPLC vials through 0.22 μm cellulose filters. The drug content in the skin was determined by HPLC measurement.

### 2.5. Effect of Nanoformulations on the Skin

#### 2.5.1. Trans-Epidermal Water Loss and Electrical Impedance

The effect of the nanoformulations on the skin was studied by the evaluation of the difference in TEWL and impedance of the intact and formulation-treated skin. For this purpose, the electrical impedance of each skin was measured, using a hand-held multimeter (LCR 4080, Voltcraft, Germany). The measurements were performed in the Franz diffusion cells according to established protocols [43,44]. After the initial hydration and equilibration, the donor compartments were filled with 500 μL of PBS. One wire of the multimeter was placed in PBS in the acceptor, the other in the donor phase and the electrical impedance was determined. The next measurement was taken after the 48-h permeation experiment. When the donor samples (300 µL) were removed, they were replaced with 500 μL PBS and the electrical impedance was re-measured.

The TEWL measurements were also carried out according to established protocols [45,46]. The skin samples were mounted into Franz diffusion cells, with the acceptor compartments filled with PBS pH 7.4 and placed in a water bath at 32 °C to hydrate and equilibrate. At first, the base TEWL values of untreated skin samples were measured. Each of the Franz diffusion cells was removed from the water bath, dried completely with compressed air, and inserted into a holder. Then the TEWL probe (AquaFluxTM, Model AF200, London, UK) was placed approximately 1.0 cm above the skin surface. The measuring time was on average 40–100 s. The steady-state values of water molecules flux density [g·m^−2^·h^−1^] were noted at comparable ambient conditions (24–26 °C and relative humidity 35–46%). After this base TEWL value measurement, donor samples were applied, cells were then incubated at 32 °C. Donor samples were removed after a permeation experiment and the skin surface was washed with distilled water, gently blotted dry with a cotton swab, and dried completely with compressed air, and TEWL was measured again.

#### 2.5.2. Fourier Transform Infrared Spectroscopy

The influence of various nanoparticulate formulations on the skin integrity was also observed using an FTIR measurement of control and formulation-treated skin. At first, spectra of intact skin were obtained on the ATR crystal (for instrumentation see Section 2.3.3). After the treatment time with the nanoformulation (24 h), the skin was washed by PBS, the permeated skin area was cut out and the samples were measured again by FTIR. Spectra of intact and formulation-treated skin were then evaluated in OriginPro software by the Fit Peak Pro function using a peak fitting procedure. Gaussian–Lorentz cross-functions were used for the best peak position and shape fit result.

#### 2.5.3. Reversibility of the Effect

The effects of the nanoformulations on the skin were studied by the evaluation of the difference in TEWL of the intact and formulation-treated skin. For this experiment, the skin samples were mounted into Franz diffusion cells, the acceptor compartments were filled with PBS at pH 7.4 and kept at 32 °C for 1 h to hydrate and equilibrate. As the first step, the baseline TEWL values of untreated skin samples were measured. Then, 300 μL of nanoformulations (empty LNC, ETZ, and PNP) were applied into the donor compartment. The donor compartment filled with PBS served as the control. The donor samples were removed after 24 h and the skin surface was washed with PBS, gently blotted dry, and thoroughly dried with compressed air. After the donor samples were removed, TEWL was measured at predetermined intervals for the next 24 h.

#### 2.5.4. Confocal Microscopy

The location of nanoparticles after the application onto the skin was studied using confocal laser microscopy (Olympus Fluoview FV1000, using UPLFLN 10× objective, Japan). For this experiment, fluorescently labelled nanoparticles (Section 2.2.4) were used. At first, 10 µm cross sections of the porcine skin after a 24-h application of fluorescein-labelled formulations were prepared using cryostat (Leica CM1850, Leica Biosystems, Deer Park, TX, USA). For the imaging, the sample was illuminated with a laser with an excitation wavelength of 488 nm, and the emission at 490 nm was recorded.

### 2.6. High-Performance Liquid Chromatography (HPLC) Analysis

Samples were measured on the HPLC instrument Prominence LC-20 (Shimadzu, Tokyo, Japan) equipped with the following: LC-20AD solvent delivery module with DGU-20A degasser; SIL-20AC autosampler; CTO-20AC column oven; SPD-M20A UV/VIS photodiode array detector; and CBM-20A communication module. LCsolution software (1.11 SP1) was used to evaluate the data. HC samples were analyzed on Kinetex^®^ column: 150 × 4.6 mm, 5 µm, RP C18, 100 Å (Phenomenex, Torrance, CA, USA). A guard column (Phenomenex, Torrance, CA, USA) with SecurityGuardTM cartridges (C18 4 × 3 mm ID, Phenomenex, Torrance, CA, USA) was used to protect the column from the skin residue. The mobile phase consisted of acetonitrile:water 32:68 (*v*/*v*); the flow rate was 0.8 mL/min; the column temperature was 25 °C; the injection volume was 20 µL; the wavelength was 245 nm; and the retention time of HC was 4.4 ± 0.1 min. HCB was also analyzed on a Kinetex^®^ column: 150 × 4.6 mm, 5 µm, RP C18, 100 Å equipped with a guard column holding the SecurityGuardTM cartridges. The mobile phase consisted of methanol:water:acetic acid 60:30:1 (*v*/*v*/*v*); the flow rate was 1.2 mL/min; the column temperature was 25 °C; the injection volume was 20 µL; the wavelength was 254 nm; and the retention time of HCB was 3.9 ± 0.1 min. Both methods were previously validated for linearity and accuracy in the range of 0.8–50 µg/mL.

### 2.7. Data Evaluation

To evaluate the permeation experiment, a calibration curve from a line of standards was constructed and the exact concentration of the drug in the samples was determined. The concentration of the active was then converted to its mass in the acceptor phase. Afterwards a graph showing a dependence of cumulative amount (Q) on time was plotted. A flux (J) of the drug was calculated as the slope of the linear part of this dependence.

All results were statistically evaluated for the outlier values by the Grubbs test in Origin Pro software. The data are presented as the mean value ± standard error of the mean (SEM).

## 3. Results and Discussion

### 3.1. Nanoparticulate Systems Characterization

#### 3.1.1. Colloidal Properties

##### Lipid Nanocapsules (LNC)

The colloidal stability of LNC was observed for 3 months after the preparation. For HC, the initial size (hydrodynamic diameter) of nanoparticles was 80 nm on average with PDI of 0.1 (Figure 1). In time this size did not deviate for more than 5 nm. The PDI of the samples stayed well below the benchmark of 0.1 signifying a great uniformity in prepared nanoparticles. The ZP of this nanosystem was around −2 mV over the whole observed time. Typically, the zeta potential of nanosystem represents its probability of stability. It is commonly accepted that the bigger the absolute values are, the more stable the system will be [47,48]. However, the charge of a particle is predominantly formed by the functional groups on the particle surface [35,49]. Therefore, in LNC, the outer shell is formed by the non-ionic PEGylated surfactant (in this case Kolliphor HS15) with prevalence of hydroxyl groups. These groups cause the overall charge to be very small with values around 0 mV. Nevertheless, here, and also in previous work [8] we present the proof of high long-term stability of LNC, even though the absolute values of ZP are low.

For HCB, the case was really similar to HC. The initial size of nanoparticles containing HCB was around 80 nm with PDI of 0.05. Again, pointing to the high uniformity of the particle size. The size of HCB-LNC did not change substantially in the following 3 months. Moreover, the PDI values never exceeded 0.05, meaning that this type of carrier is very suitable for drugs like HCB. It can mainly be thanks to its higher lipophilicity and higher affinity to the oil core. The ZP values are nearly identical to those of HC-LNC, which was expected as the incorporated drug does not influence the surface charge of the nanovector.

##### Polymeric Nanoparticles (PNP)

Due to the limited stability of HCB-PNP, its original composition, which was the same as for HC-PNP, was slightly changed to achieve an optimal formulation. For polymeric nanoparticles, the main mechanism of API-polymer entrapment are the hydrophobic and hydrophilic interactions [50]. For this reason, HC, which is less lipophilic, can be held in the nanoparticle by the combination of hydrophilic and hydrophobic interactions. In contrast, more lipophilic HCB is limited to hydrophobic interactions with the polymer. Based on these reasons, a higher amount of PLGA, and consequently also PVA for maintaining a stable formulation, was used for encapsulation of HCB into the PNP. The PNP were generally stable for 1 week in the aqueous medium at 4 °C in the dark. Subsequently, the formulation degraded, and HC/HCB crystalized. Due to low stability of the nanosystem in aqueous medium, PNPs can be freeze dried for the stability prolongation (see Appendix A) [51].

The particle sizes, PDI and ZP are shown in Figure 1. Based on the very low PDI, both PNP created very uniform systems. The size was between 180 and 200 nm, which is typical for PLGA nanoparticles [52]. The HCB-PNP were about 20 nm larger compared to HC-PNP. This could be caused by the higher viscosity of aqueous phase as a result of higher PVA amount in the formulation [53]. The ZP values were between −15 and −7 mV for HC-PNP and HCB-PNP, respectively. The lower absolute values of HCB-PNP were given by the higher PVA concentration in the formulation [54].

##### Ethosomes (ETZ)

The stability study of ethosomes was again carried out by storing the samples at laboratory temperature (25 °C) for 60 days and observing the development of particle size and PDI (Figure 1). As was mentioned above, ZP is an important physical parameter for predicting the stability even for ETZ [55]. Our prepared ethosomal formulation showed a negative zeta potential as expected due to the presence of ethanol, which causes a negative charge; however, the absolute values of ZP were low, reaching only units of mV. Such a charge is not enough to prevent the complete separation and repulsion of the nanoparticles [56], therefore certain limits to the colloidal stability were expected. This was confirmed by the development of a particle size study. The initial size of HC and HCB loaded ETZ differs by circa 100 nm, with the latter being smaller with a size of around 150 nm. A peculiar trend can be seen in the particle size development during the following storage. HC-ETZ size gradually decreased until it reached 180 nm in the second month after preparation. Conversely, HCB-ETZ particle size increased from an initial 150 nm to 270 nm. This can be due to relatively higher polydispersity and lamellarity variance of ETZ. Even our ETZ are prepared by a low-energy method; therefore, upon storage, the system tends to re-distribute into more thermodynamically favourable state than it was prepared in. Our ETZ are also not processed further by eg. high-pressure or high-sheer homogenization, so the particles are predominantly multilamellar, fused or combined and all that can lead to the particle size changes in time. The development could even be connected with the aggregation behaviour of the samples (see Section 3.1.4). In general, this stability study confirmed that HC and HCB loaded ETZ were stable for two months and, considering their size, a suitable formulation for topical application. For drug delivery to the skin, an optimum ETZ size is considered to be less than 200 nm, although vesicles smaller than 300 nm are considered to be efficient enough in localizing the drug deep into the skin [57].

##### Comparison of the Colloidal Properties of the Three Systems

Analyzing the colloidal properties of the three studied systems, the following findings can be concluded. Of the three systems, LNC showed the smallest size of around 80 nm and a low PDI (0.1). The zeta potential was very low (around zero); however, this value did not relate to any instability of the samples, which did not change their properties for at least three months. Compared to LNCs, PNP were larger (around 200 nm) and of uniform size. Although they showed higher absolute values of ZP, their stability was low (only one week) and should be supported by other processes (e.g., freeze-drying). ETZ showed their size of around 200 nm and higher polydispersity than LNC and PNP. They were stable for two months, but during this time, their size developed probably to reach a more energetically preferred state. Besides comparing the particulate properties, an interesting finding was no relationship between the zeta-potential absolute values and stability of the systems.

#### 3.1.2. Encapsulation Efficiency and Drug Load

The nanoparticulate systems in this work were designed to contain 0.05 or 0.1 wt% of HC or HCB in the total formulation. Nevertheless, each system has a different capability of incorporating the introduced amount of the API. Therefore, it is important to measure the encapsulation efficiency (EE) and drug load (DL). The rate of drug encapsulation influences the release rate of the drug from the nanoparticle and consequently it can be an important variable in the skin permeation behaviour.

HC is a lipophilic compound with low aqueous solubility (logP = 1.3, according to Chemicalize software, Chemaxon, Hungary), which favours its incorporation into lipophilic nanoparticles, such as LNC or ETZ. HC solubility was higher in IPM (core oil of LNCs) than in PBS (see Appendix A) and, moreover, the solubility was even higher in the LNC lipid phase, suggesting the probability of high EE rates. The hypothesis was confirmed as the EE of HC in LNC was established to be 80 ± 2% (DL = 1.5 ± 0.2%), which is comparable or even better than other developed formulations for HC [22,23]. The PNP were able to incorporate only a limited amount of HC compared to LNC (EE = 27 ± 2%, DL = 2 ± 0.5%). Establishing the EE in a liposomal formulation is often problematic and may produce uncertainties in the methodology and computations. In case of our ETZ, the EE value of HC was determined close to zero, indicating that almost all HC in the formulation was dissolved in the aqueous phase and only a negligible amount was inside the phospholipid membranes. This is in accordance with the high solubility of HC in the ethanol/water mixture (see Appendix A). The measured low EE value, however, reflects only the solubility in the aqueous phase in general but not the real amount of HC incorporated inside the vesicles in the water phase. Therefore, we suppose that in reality, the EE value for ETZ can be higher.

HCB possesses higher lipophilicity (logP = 2.9, according to Chemicalize software, Chemaxon, Hungary) and around ten times lower aqueous solubility than HC itself. This elevates the probability of HCB incorporation into suitable nanosystems. The solubility of HCB in IPM and the LNC lipid phase was quite high (see Appendix A) and it also influenced the encapsulation rate, which reached 86 ± 1% with DL = 1.8 ± 0.3%. Although, for PNP formulation, the EE reached very high values of 78 ± 1% (DL = 3 ± 0.6%), it was not possible to prepare stable final formulation with 0.1% HCB (due to instantaneous HCB crystallization during sample preparation). Therefore, the formulation with HCB contained only 0.05% of HCB. EE in HCB-ETZ formulation was again close to zero. We explain this effect in a similar way as for HC (see above).

#### 3.1.3. FTIR of Nanosystems

To better understand the level of interaction of the drugs and nanoformulations on a molecular level, the HC or HCB loaded systems were studied by ATR-FTIR. This method is quite useful to preliminary characterize nanosystems, as it can give an idea about the release behaviour of the drug from certain formulations [42]. The main question here was whether there would be any difference in the spectra of the pure drug, drug loaded nanosystem, or drug-component physical mixtures. Figure 2 presents the FTIR spectra for HC (upper row) and HCB (bottom row). The spectrum of pure HC shows two notable peak groups between 1600 and 1750 cm^−1^ as one triplet at 1610, 1630, and 1640 cm^−1^ and a second less intensive doublet at 1705 and 1715 cm^−1^. When the HC was mixed with PLGA and PL in a physical mixture, peaks from both components were still visible as these two compounds did not mix together at the molecular level. When HC was combined with IPM, the HC peaks between 1600–1750 cm^−1^ were no longer visible, even in a simple physical mixture. This can be ascribed to the fact that HC was solvated with the IPM and therefore the peaks of pure drug were obscured and only those of the solvent were observed by FTIR. The spectra of HC-loaded LNCs, PNPs, and ETZs hinted to the level of HC incorporation into the nanoparticles and correlated with findings of encapsulation efficiency of those nanosystems. For HC-LNC, the HC peaks of interest were practically gone as opposed to PNP and ETZ, where they could still be seen to certain level, in ETZ even more intensively than in PNP.

For HCB, the trend was very similar to HC. This time, the area of interest was again between 1600–1750 cm^−1^, but with distinct positions for HCB. The middle peak of the triplet at 1630 cm^−1^ was here dimmed and the respective positions were: 1620, 1630, and 1650 cm^−1^. On the other hand, the doublet was more intensive and its peak positions for HCB were 1715 and 1735 cm^−1^. Even for HCB when it was formulated into LNCs, these groups of peaks were not visible, which was pointing to a high level of drug incorporation into the nanoparticle body, which was in accordance to high EE (>85%), as shown in the previous chapter. As PNP and ETZ showed much lower EE for HCB, this fact was shown in their individual spectra. In both cases, HCB-PNP and HCB-ETZ contained bands from both the drug and formulation components.

The fact that both HC and HCB are almost totally incorporated into LNCs, but not into the other two nanoformulations, can determine the release behaviour in subsequent in vitro (Appendix A) and ex vivo experiments and decide the effectivity of the delivery systems.

#### 3.1.4. Transmission Electron Microscopy

The observation of the nanosystems by TEM confirmed the predicted particle size obtained from the DLS measurements. It also elucidated the morphology of each individual system (Figure 3). LNC were shown as droplet-like nanoparticles with a slightly variable size. This type of LNC system contains more of the core oil (IPM) than previously studied formulations [8], therefore the inner compartment is more visible, and the observed morphology can really be assimilated to the oil droplet with semi-solid surfactant stabilized walls, as described in the LNC definition [34]. Polymeric PLGA nanoparticles are uniform round nanospheres with a narrow distribution of sizes. PLGA particles generally form similarly shaped objects, as is reported in several research articles [58,59]. The photograph of ETZ shows nicely uniformly shaped liposomal particles. It also provides the proof of ETZ aggregation, which is quite usual for phospholipid vesicles [55]. This fact can also hinder the DLS measurements, and it is necessary to pay attention to proper sample preparation and homogenization prior to the analysis. Otherwise, the reported size can be falsely determined to be larger than the real one and the sample could be inappropriately eliminated from the study.

#### 3.1.5. Final Nanoparticulate Systems for Ex Vivo Experiments

Collectively, three distinct nanoformulations were prepared according to established procedures. The summary of their colloidal characteristics is stated in Table 4, where the reader is able to compare the individual differences in particle size (hydrodynamic diameter) and EE of each system as they were subsequently applied in release and permeation studies. As the particle size is concerned, all of the types contained particles sized between 80 and 300 nm, positioning them in between other formulations developed for corticoids delivery [60,61,62,63]. Additionally, as described in the work of Lademann et al. [64], this particle size is suitable for topical delivery with possible enhancement via follicular targeting or transcutaneous penetration. The EE of presented systems varied from lower values for PNP of less than 30% up to almost 90% for LNC. The rate widely depended on the nature of the incorporated drug and the used excipients and could have extensive influence on the permeation characteristics of the drug in question.

### 3.2. Ex Vivo Permeation Study

#### 3.2.1. Hydrocortisone

In order to assess the efficacy of various nanoformulations in (trans-)dermal delivery of corticosteroids, we conducted a series of ex vivo permeation experiments on both full-thickness porcine and human skin. Figure 4 shows the permeation characteristics of hydrocortisone (HC) delivered from LNC, PNP, ETZ, and PBS suspensions for 48 h. It is clear from the graph in Figure 4a that a simple suspension was very ineffective in the transdermal delivery of HC. Nevertheless, thanks to the more lipophilic environment of the skin, a certain amount of the drug permeated into the acceptor phase. Generally, for the common application of topical corticosteroids, the idea is to concentrate most of the drug into skin layers and minimize the transdermal transport. It is especially crucial for a long-term therapy with larger drug dosage where there is the danger of systemic side effects caused by prolonged exposure to larger amount of the corticoid [65]. It could be said that this minimization was achieved by the application of an HC suspension. However, even if the passage was not very intensive, the formulation as a whole was very ineffective and material consuming. The total permeated amount of HC from the 1% PBS suspension was 1/1000 of the total applied dose, which was certainly not in favour of the cost-reasonable production of a formulation. The transdermal deliveries of HC from LNC and PNP were very similar to each other. Compared to the suspension, the delivery was around four times higher. On the other hand, HC ETZ were able to enhance the passage of the drug quite extensively, as HC delivered from ETZ reached around a 15 times higher cumulative amount than the delivery from suspension control.

The parallel experiment on full-thickness human skin (Figure 4, right column) resulted in similar trends. Here, the permeation of HC from the PBS control was totally diminished, and no drug could be analyzed in the acceptor medium. Due to the exceptional barrier function of human skin, which is even more rigid and intricate than the porcine alternative [66], the delivery of HC from LNC and PNP was very low. Again, they are quite similar to each other; however, only units of the drug were able to pass through the skin. Opposite to this, ETZ, even on the human skin, showed their great potential as transdermal enhancing formulation. The amount of HC that permeated into the acceptor phase from the HC-ETZ formulation was only slightly lower than the one found for the porcine skin. ETZ seem to influence the stratum barrier of the skin [67], which would result in similar permeation values. When the one difference between porcine and human skin, which is the incredibly complex composition of human SC lipid matrix, is taken away, the remaining barrier of the dermis and subcutaneous tissue is quite similar in both skin types [68]. Therefore, we could observe a bit slower, but in the end, a comparably intensive permeation of HC from ETZ through the porcine and human skin occurred.

The dermal entrapment and concentration of the corticoid is the most desirable outcome for successful skin disease therapy. Figure 4e shows the amount of HC that was quantified from the drug-permeated porcine skin biopsies. Again, the least effective entrapment was achieved from the PBS suspension formulation. Compared to this, PNP were able to target HC into skin layers more effectively. However, LNCs managed even better outcomes with twice the HC delivered than from the suspension (32 µg/g compared to 16 µg/g). ETZ, as expected from previous transdermal results, opened the skin for the drug passage and created the environment for HC accumulation in the skin layers. The ethanolic solution in ETZ composition favourably influences the solubility of HC, and as ETZ affect the skin for certain time, they could also carry over this effect to the skin environment and promote the detention of HC [69]. The situation in the human skin was again very similar to porcine skin. Overall, the absolute amount of drug retention was lower than in the previous case with PBS suspension being the least effective formulation, followed by PNP, LNC, and then ETZ. On top of that, in this case, LNC and PNP were able to ensure 10 times higher HC accumulation in the skin compared to the permeated amount. ETZ were the most effective formulation with a comparable amount of HC entrapped in the human skin.

#### 3.2.2. Hydrocortisone-17-Butyrate

Hydrocortisone-17-butyrate being a more potent derivative of HC necessitates even more closely controlled administration and, where needed, a minimization of the unintended transdermal permeation. The higher lipophilicity of HCB dictates its fate in the skin permeation, differentiating it from HC behaviour. The permeation profiles of HCB from various formulations through the porcine skin are shown in Figure 5a. It can be noted that the total amount of the permeated drug was approximately ten times lower than it was with HC (Figure 4a). This fact is the result of the high lipophilicity and low aqueous solubility of the drug (Appendix A), causing it to remain in the lipophilic environment of the skin and diminishing the permeation into the acceptor phase. The transdermal permeation from LNCs was totally under the detection limit. No HCB was analysed from the acceptor phase in any sampling time point. This is exactly the same as the example where HCB is delivered from a lipid formulation through lipophilic skin regions into a highly unfavourable PBS environment. Compared to this, when HCB was delivered from PBS and PNP, which both possess water as the main solute, the transdermal permeation was present, nevertheless, very weak. Moreover, the permeated amount was far from sufficient to cause any unwanted effects in the system. Therefore, the dermal delivery of HCB from PNP could be considered safe in regard to the systemic effects. ETZ again showed their capability to deliver drugs through the skin barrier in a very efficient way [25]. Even though the total permeation was in units of micrograms, it was the most intensive one from the tested formulations. The situation in human skin corresponds to previous findings and the generally known fact of a stronger barrier function compared to animal models. The passage of HCB from LNC, PNP, and PBS suspension through the human skin was practically zero and often at the limit of quantification of the HPLC method. The liberation of HCB from ETZ achieved almost the same cumulative amount through the human skin as through the porcine skin (3.2 µg/cm^2^ and 4.8 µg/cm^2^, respectively). Pointing again to the hypothesis that ETZ significantly fluidize the skin barrier and the drug has to pass only the dermis and subcutis tissue. A similar observation was also reported in the literature, when Feldman et al. [68] studied corticosteroid permeation through normal and SC removed human skin in vivo. The group postulated that by removing the SC barrier, the penetration of the drug doubles, which corresponds to our findings and could explain why an ethosomal formulation produced such similar results in all studied cases.

As the HCB dermal entrapment is concerned, the situation in porcine skin is analogous to HC. Albeit the delivered amount is lower than the one of HC, the effectivities of studied formulations are in the same order: PBS < PNP < LNC < ETZ. On the other hand, the case of human skin revealed exceptional results. The amount of HCB entrapped in the human skin was overall considerably high for all of the formulations with much smaller differences between each other than observed before. In this experiment, the LNCs were almost as comparably powerful as the ethosomal formulation. The amounts of HCB left in the skin layers after the exposition were 25 and 29 µg/g for LNC and ETZ, respectively. The accumulation of PNPs was slightly lower than in previous formulations; however, it must be kept in mind that PNP contained half the amount of HCB as LNC and ETZ (0.05% HCB in formulation compared to 0.1% in LNC and ETZ). Therefore, the relative effectivity is not lagging behind the ones of their competitors.

### 3.3. Effect of the Nanoformulations on the Skin Barrier

#### 3.3.1. TEWL and Skin Electrical Impedance Measurements

In order to see the effects that the various nanoformulations have on the skin barrier, skin electrical impedance and TEWL were measured before and after the exposition to the formulations. Figure 6 depicts the ratios of these values for porcine skin (Figure 6a,c) and for human skin (Figure 6b,d). The effect was observed for all three nanoformulations and for PBS. The skin treated by PBS was set as a control and therefore, all ratio values for PBS-treated skin were set to 1, according to protocol described previously [70].

TEWL is generally used as an indicator of the state and barrier function of the skin. The flow of water molecules through this barrier is tightly regulated and is proportional to lipid composition and arrangement in the stratum corneum. The average level of TEWL for human skin is considered to be between 1–25 g·m^−2^·h^−1^ [71]. Any pathophysiological or mechanical changes to the skin result in elevated values of TEWL, with this change being able to foreshadow the level of barrier disruption [72,73]. Therefore, in this experiment the ratio of the TEWL values yields a number lower than 1, and the lower the ratio is, the larger the effect of the nanoformulation. As can be seen in Figure 6a,b, the lowest TEWL ratio was measured for the skin exposed to the ETZ formulation. Both LNC and PNP also showed a slight effect on the skin barrier; however, compared to ETZ it was negligible. Overall, the observed trend in the formulation effect on the skin barrier was found to be similar in both human and porcine skin, supporting the latter as a suitable ex vivo model for studies.

The electrical impedance of the skin barrier is another robust method used for a long time for the assessment of the skin barrier state [74,75]. Here, the data were handled the same way as described for TEWL, just the expected values for the ratio before and after are higher than 1 this time, because after the exposition to a formulation, the impedance value decreases, as the skin is more fluidized and open to the permeation. The results shown in Figure 6c,d confirm the findings from the TEWL part. Even here, the most significant effect on the skin was caused by the application of ETZ (e.g., the ratio value is the highest of them all). That can be due to the enhancing effect of ethanol in the formulation, which can disrupt the skin barrier more and thus enable the passage of API through the skin more effectively than other formulations. Although the reversal was not as fast and clear as for other formulations, the incorporation of ethanol in ethosomes showed lower effects on the barrier properties than even traditional and novel permeation enhancers [44]. LNC and PNP showed levels of alternation comparable to the PBS.

#### 3.3.2. Fourier-Transform Infrared Spectroscopy

Another way to further study the influence of nanoformulations on the skin was to observe the difference in significant peak positions between the intact and formulation-treated skin. The band assignment can be found in Appendix A. The main regions of interest were the CH_2_ symmetric (ʋ_s_CH_2_) and asymmetric (ʋ_as_CH_2_) stretching vibrations at around 2850 and 2918 cm^−1^, respectively. These bands are significant indicators of the ordering level of lipid bilayers in the SC lipid matrix. For healthy intact porcine and human skin, the position of those bands is reported to be at 2850, 2918, and 2850, 2919 cm^−1^, respectively [76]. By an induced disruption, the peak positions shift to higher wavenumbers, signifying the loosening of the hydrocarbon chains in the lipid matrix [77]. The application of nanoformulations on both porcine and human skin influenced not only the position of the peak, but also its intensity and shape. The peaks of intact skin possessed a very sharp distinct shape with maximum at given position. The application of nanosystems, namely ETZ, affected the peak position by a shift in the ʋ_as_CH_2_ vibrations by ~ 3 cm^−1^ in porcine and 5 cm^−1^ in human skin (Figure 7), signifying a significant weakening of skin barrier function. It also distorted the shape of those peaks with their signals being very weak compared to the rest of the spectra. For LNC and PNP both, the peak position and shape remained similar to the intact skin, with the change being less than 1 cm^−1^. The trends are again very similar in both skin types, confirming the similarity between those two.

#### 3.3.3. Reversibility of the Skin Barrier Function

We confirmed that all of the tested nanoformulations influence the skin barrier to some extent, some more than the others. It is not an uncommon thing to see various changes in the skin after a disruption, treatment, or application of formulations. It is however important to know whether this effect is permanent, leaving the skin in a state close to disruption, or reversible. If it is the latter, one must find out how long it takes for the skin barrier to return to its original properties. Such reversibility measurements are well established in the evaluation of permeation enhancers [44,75,78]. Nanoformulations are also a type of enhancing system; therefore, we conducted a reversibility study, where we observed the TEWL of skin treated for 24 h by all three nanosystems (ETZ, PNP, and LNC) and hydrated by PBS as a control and then measured the same in set time intervals up to 24 h after the removal of the formulation. Figure 8 depicts the development of TEWL values of porcine (Figure 8a) and human skin (Figure 8b). The values of the intact porcine skin were very uniform and spread around 25 g·m^−2^·h^−1^. The same starting values for the human skin were more scattered than those of porcine skin; nevertheless they were overall lower at around 20 g·m^−2^·h^−1^, which corresponds to a greater barrier function of the human skin. After the removal of the formulations, the porcine skin expressed elevated values of TEWL, which were almost on the same level. From our experience, TEWL values above 50 g·m^−2^·h^−1^ are at the upper borderline of what full thickness skin is able to possess and usually do not go much higher. Therefore, here in the porcine skin, the difference in the individual effect is not so noticeable. The porcine skin treated by all three types of nanoformulations did not regain its original properties in the observed 24 h after the formulation removal. The TEWL values of LNC, PNP, and PBS treated skin are albeit very close to the initial values; they did not reach the threshold of 25 g·m^−2^·h^−1^. On the other hand, the situation for the human skin was far clearer. The effect of individual formulations is distributed between values of 23 g·m^−2^·h^−1^ for LNC up to almost 60 g·m^−2^·h^−1^ caused by ETZ. Moreover, both PNP and LNC seem to affect the human skin, TEWL less than PBS itself, which can be related to other components in the formulation (e.g., lipids, polymers, etc.). The drop in TEWL after the formulation removal was very quick for all of the subjects already in the 2nd hour. LNC-, PNP-, and PBS-treated skin even reached TEWL values of the intact skin. For ETZ treated skin, the reversal of TEWL was slower and stabilized at around 30 g·m^−2^·h^−1^, which was still higher than the starting 22 g·m^−2^·h^−1^ before the application, hinting to limited recovery of the skin barrier function during the 24 h.

### 3.4. Confocal Microscopy

To deeper understand the mode of action by which each nanoformulation promotes the skin drug uptake, the localization of fluorescein-labelled nanosystems in the skin was observed by confocal microscopy. When the skin was exposed to a fluorescein sodium suspension in PBS (0.1% *w*/*v*), the signal obtained from the confocal microscopy clearly indicated a dense concentration of the fluorescent at the skin surface (Figure 9). No or only a minimal leakage into skin layers was visible and it could be assumed that such a simple formulation as the suspension was not able to promote the efficient penetration of fluorescent sodium through stratum corneum into the epidermis. The LNCs when put into contact with the skin seemed to remain mostly near the surface layer. As can be seen from the micrograph, the nanocapsules created a uniform coating but also partially infused the stratified skin layers. A weaker signal was also obtained from epidermis in the case of LNC application. It was possible that as LNC created an occlusive film at the skin surface, the skin became more hydrated and the LNC lipid components were able to penetrate into stratum corneum or even to lower layers and act there as promoting agents for drug permeation.

As the PNP mode of action is concerned, it is generally assumed that polymeric nanoparticles sized around 200–300 nm are designed for dermal delivery and particularly for follicular targeting and act as great depots for the controlled liberation of the API without a significant burst effect [79,80]. In our case, after their application on the porcine skin, FNa-labelled PNPs seemed to readily accumulate in the hair follicles (detail enlarged in Figure 9). Owing also to their relatively high dilution, PNP did not create the continual layer as observed with LNCs. Therefore, according to the literature [81], we can confirm the predicted mode of action of PLGA polymeric nanoparticles as mainly a follicular concentration and the continual liberation of the carried API.

Ethosomes as carriers were developed for quick and intensive skin passage. They should be able to take advantage of their high ethanol content and liposomal morphology and not only dissolve high API content, but also carry it through the skin layers into the blood stream in large amounts [82]. The confocal microscopy of labelled ETZs on porcine skin revealed their strict deposition in the uppermost layers of the skin. What can be also noted from the micrograph is the widening of the stratified layers. Compared to the skin treated by LNC or PNP, the barrier with ETZs seemed to be more fluidized and loosened. This fact may be ascribed to the composition of ETZ, where the high ethanol content caused elevated mobility of the lipid layers and made this barrier more open to the passage of external compounds. What was surprising was the absence of any signal in the deeper layers of the skin. From the analysis of the acceptor media from this exact experiment, we found that the NBD signal was not present; therefore, ETZ did not permeate the skin.

### 3.5. Future Perspectives of the Developed Nanosystems

The nanoparticulate systems show great potential in the topical delivery of actives for the treatment of skin diseases. The extensive research from current years supports the increasing interest in various lipid and polymeric nanovectors containing not only corticosteroids, but also many more actives for (trans)dermal delivery [83,84,85]. However, there are rather few studies comparing the efficiency of different types of nanosystems for specific drugs. In this study we aimed at such comparisons of LNC, PNP, and ETZ. Although they are all defined as nanoparticulate colloidal systems, they possess very different properties, including composition, size, etc. This makes it difficult to establish clear relationships between the nanoparticulate properties and effectivity. However, thanks to the comparison of the nanosystems subjected to the same experimental design, it was possible to evaluate their suitability for dermal and/or transdermal applications. This should be considered as the main contribution of this study.

As confirmed by the ex vivo experiments, all of the nanoparticulate systems came out as potent medium for topical application of HC and HCB. Particularly ETZ were able to significantly increase the flow of both to the skin tissue but also further to the acceptor phase. Moreover, ETZ have rather disturbing effects on the skin barrier, which did not recover completely after their application. This fact can be limiting for their further use for treating skin diseases related to impaired skin barriers. On the other hand, ETZ can be excellent in the transdermal delivery of drugs applied on healthy skin [32,86,87].

Considering the intended application of corticoids requiring a maximum local concentration in the skin but minimum systemic absorption, the LNC and PNP showed more favourable effects. They delivered substantial drug amounts into the skin with almost no transdermal passage, especially for HCB. This drug shows higher lipophilicity compared to HC, which is a reason for its higher drug loads in the lipophilic nanoparticles. Probably this can also be the reason for the higher accumulation of HCB in the skin. Moreover, the more lipophilic drug does not prefer to leave the skin tissue and enter the more hydrophilic acceptor.

The observed results can be put into the context of other literature where the authors studied the topical application of corticosteroids. Even though the majority of results are obtained in mouse or rat skin, the relationship between the permeability of the skin model membranes is well described [88,89]; therefore, it could be extrapolated for certain trend observations and comparisons. Compared to the study conducted by Kim et al. [22] with a liposomal formulation of HC, the observed unwanted transdermal permeation of the drug was significantly lessened in our case. Moreover, liposomes were not able to ensure a high concentration of HC in the skin layers. In the work of Laugel et al. [90], where HC was applied in a form of a simple or combined emulsion, the difference between transdermally permeated and dermally entrapped HC in rat skin is not as evident as, for example, the results obtained in all of our nanoformulations, highlighting the importance of suitable nanoformulations in topical delivery. In his research, Mombeiny et al. [25] studied the skin permeation of HCB from the ethosomal formulation. The cumulative amount of permeated HCB was found to be elevated when delivered from the nanovector and even more when combined with electrophoresis. Unfortunately, the important dermal accumulation is not reported.

## 4. Conclusions

This work offers a comprehensive comparison between three nanoparticulate systems of different composition and morphology used for the encapsulation and topical administration of two commonly used topical corticosteroids. Hydrocortisone (HC) and hydrocortisone-17-butyrate (HCB) were formulated into LNC, PNP, and ETZ. The systems were tailored with respect to their high drug content and colloidal stability. We were able to reach 0.1% of HC or HCB in all formulations, except for PNP containing 0.05% HCB.

In an ex vivo permeation study, the formulations showed high effectivity in the transdermal permeation (especially ETZ) and dermal accumulation (PNP and LNC). The observed results corresponded to effects of the particular formulations on the skin barrier as studied by FTIR, TEWL, and confocal microscopy. Opposite to LNC and PNP, ETZ significantly affected the arrangement of the lipids in the SC by increasing their fluidity. This effect was so strong that even the reversibility of the skin barrier function was much slower than for the LNC- and PNP-treated skin. Each of the studied nanosystems interacted with the skin barrier by unique mechanisms. The PNP were found to readily accumulate in the hair follicles, creating a depot for the sustained delivery of the drug. On the other hand, LNC promoted the API permeation by an occlusive effect at the skin surface together with permeation of their lipid components into the epidermis. ETZ showed remarkable effects on the fluidization of the stratified layers of the skin barrier.

Each of the presented formulations have relevant positions in the topical delivery of drugs for the treatment of skin conditions. LNC and PNP show the potential for the effective dermal delivery of the drug of choice, intensifying the administered treatment. Additionally, the limited transdermal permeation of the LNC provided could ensure the minimization of the side effects of highly potent anti-inflammatory actives. On the other hand, if more intensive systematic treatment would be necessary, ETZ can ensure sustained high drug flux through the skin barrier. In conclusion, according to the desired aim of drug application, the nanosystem can be chosen and tailored for the specific API and serve as an effective dermal or transdermal carrier.

## Figures and Tables

**Figure 1 pharmaceutics-15-00513-f001:**
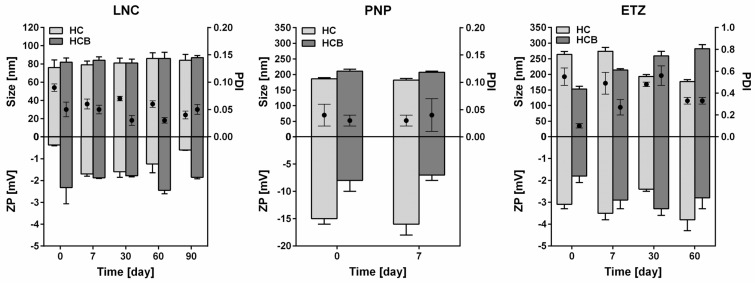
Particle size as the hydrodynamic diameter (bars), polydispersity index (PDI, scatter), and zeta potential (ZP, negative bars) of lipid nanocapsules (LNC), polymeric nanoparticles (PNP), and ethosomes (ETZ) containing 0.1% of hydrocortisone (HC) and hydrocortisone-17-butyrate (HCB; HCB-PNP contained 0.05% of HCB) measured on the day of preparation and consequently for following days. LNC and ETZ samples were stored at 25 °C. PNP were stored at 4 °C. *n* ≥ 6.

**Figure 2 pharmaceutics-15-00513-f002:**
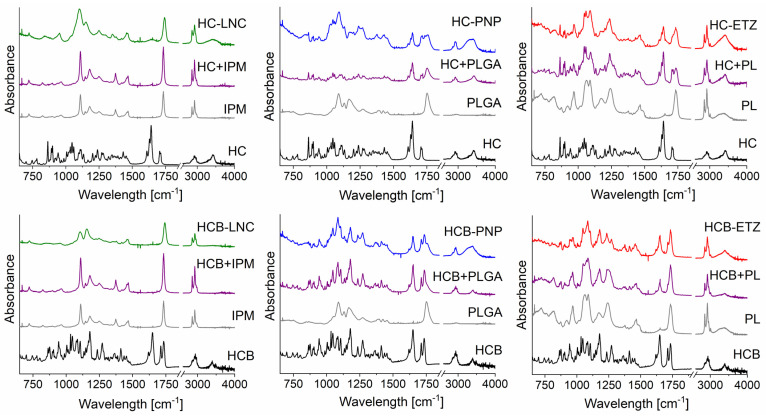
FTIR spectra of pure API-black (named HC; HCB), physical mixture of API with individual components of the nanoformulations—purple: isopropyl myristate (named HC/HCB + IPM), poly(lactic-co-glycolic acid) (named HC/HCB + PLGA) and Phospholipon 90G (named HC/HCB + PL); and of the freeze-dried API nanoformulation: green—lipid nanocapsules (HC/HCB-LNC), blue—polymeric nanoparticles (HC/HCB-PNP), red—ethosomes (HC/HCB-ETZ).

**Figure 3 pharmaceutics-15-00513-f003:**
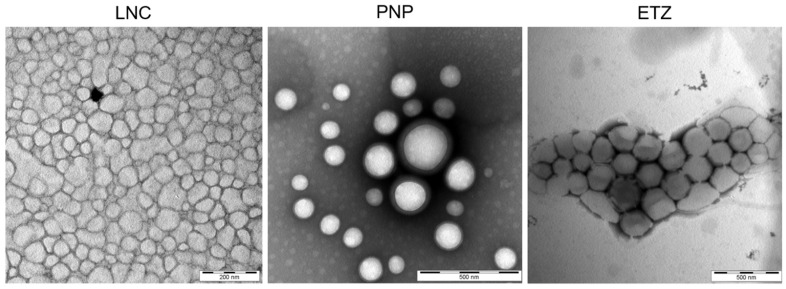
Transmission electron microscopy of HC loaded-nanosystems studied in this work: lipid nanocapsules (LNC), polymeric nanoparticles (PNP) and ethosomes (ETZ).

**Figure 4 pharmaceutics-15-00513-f004:**
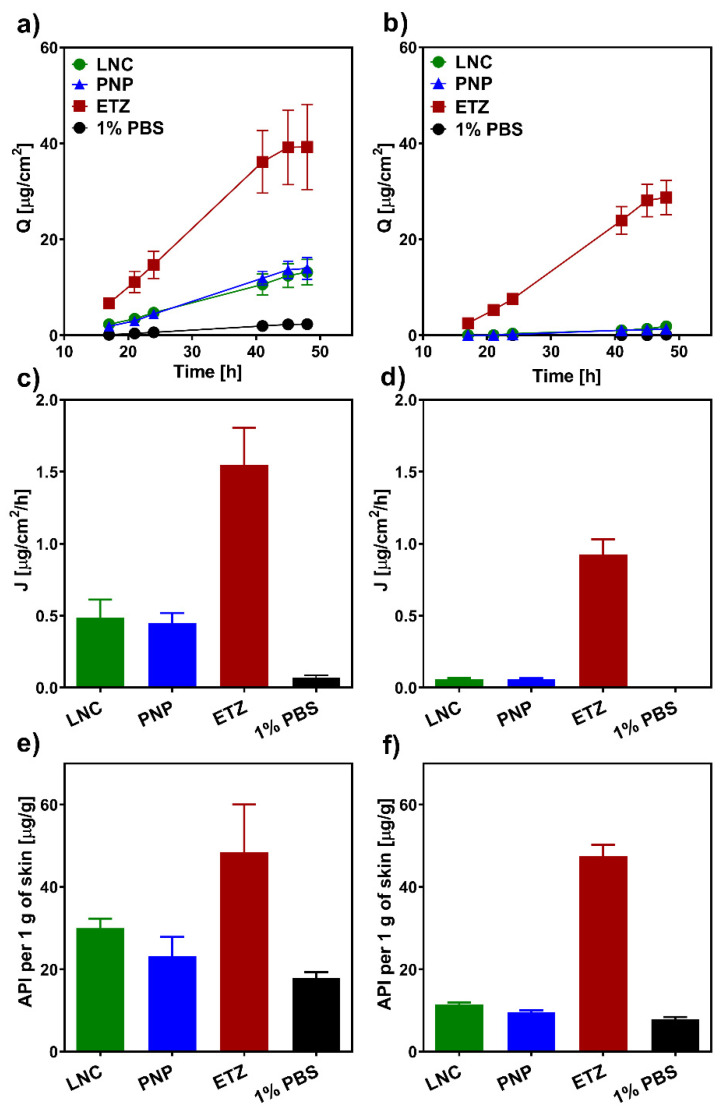
Permeation characteristics of hydrocortisone from the ex vivo permeation study. The left column (graph (**a**,**c**,**e**)) represents the results obtained on porcine skin, the right column (graph (**b**,**d**,**f**)) contains results from the human skin. The first row (graph (**a**,**b**)) depicts permeation profiles as the dependency of cumulative amount on time. The second row (graphs (**c**,**d**)) contains flux values of HC liberated from various nanoformulations. The bottom row (graph (**e**,**f**)) shows the entrapment of HC in the skin presented as the amount of HC per 1 g of the skin. *n* ≥ 12.

**Figure 5 pharmaceutics-15-00513-f005:**
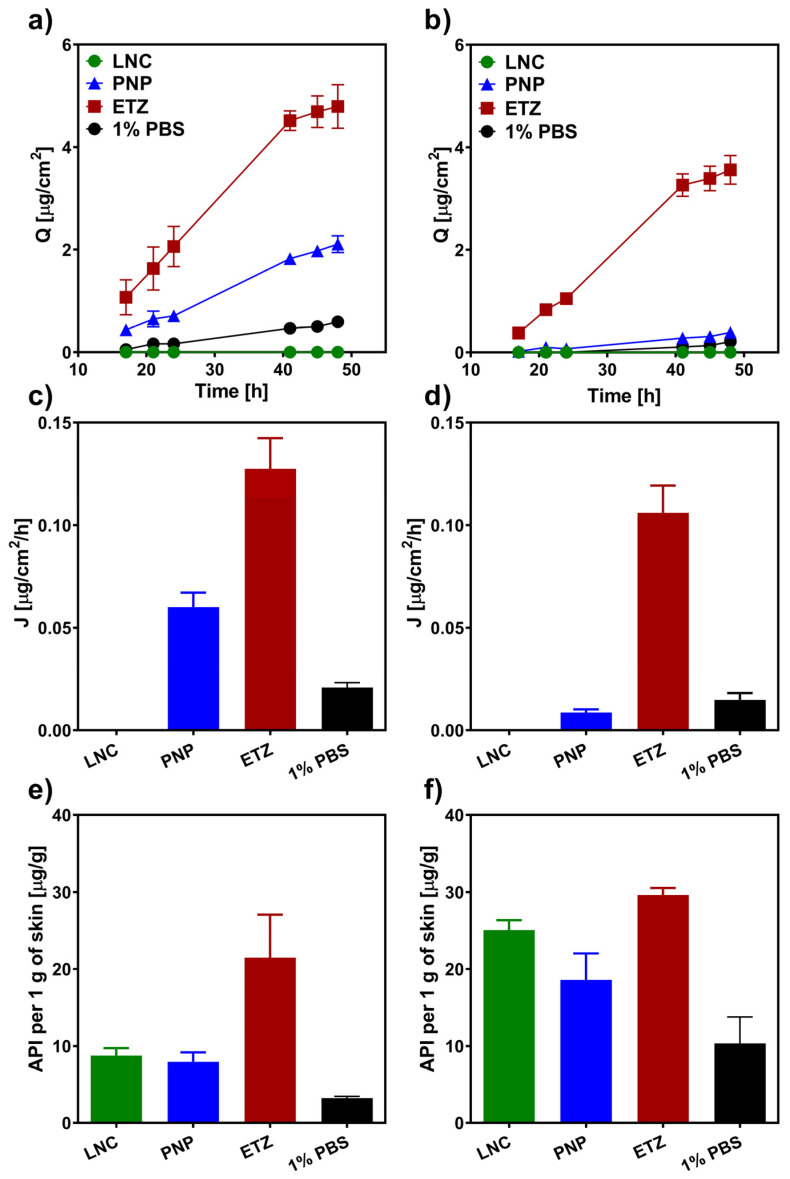
Permeation characteristics of hydrocortisone-17-butyrate (HCB) from the ex vivo permeation study. The left column (graph (**a**,**c**,**e**)) represents the results obtained on porcine skin, the right column (graph (**b**,**d**,**f**)) contains results from the human skin. The first row (graph (**a**,**b**)) depicts permeation profiles as the dependency of cumulative amount on time. The second row (graphs (**c**,**d**)) contains flux values of HCB liberated from various nanoformulations. The bottom row (graph (**e**,**f**)) shows the entrapment of in the skin presented as the amount of HCB per 1 g of the skin. *n* ≥ 12.

**Figure 6 pharmaceutics-15-00513-f006:**
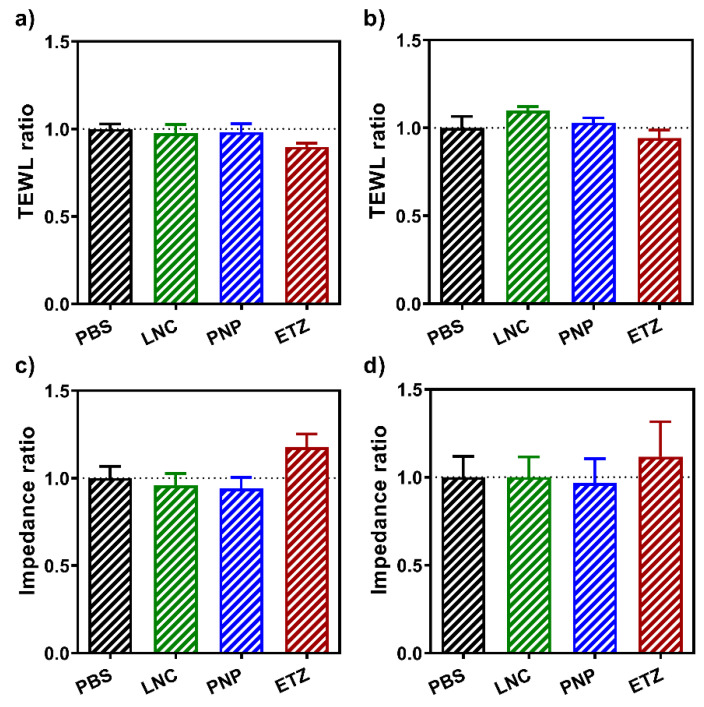
TEWL and impedance ratios of values obtained before and after the application of the respective nanoformulation (LNC, ETZ, PNP) or control (PBS) on (**a,c**) the porcine skin, (**b,d**) the human skin. *n* ≥ 8.

**Figure 7 pharmaceutics-15-00513-f007:**
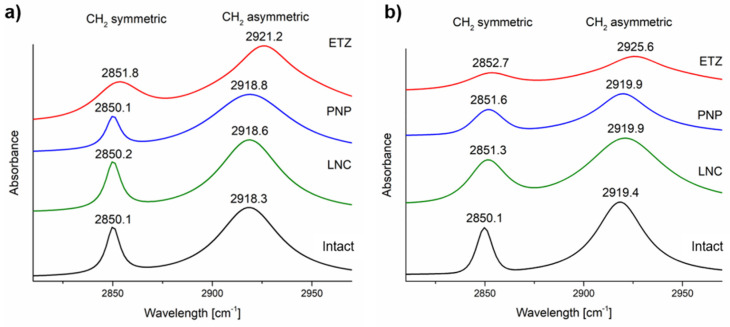
Fitted significant peaks of CH2 symmetric and asymmetric stretching from the ATR-FTIR spectra of (**a**) porcine, (**b**) human skin treated by lipid nanocapsules (LNC), polymeric nanoparticles (PNP), and ethosomes (ETZ).

**Figure 8 pharmaceutics-15-00513-f008:**
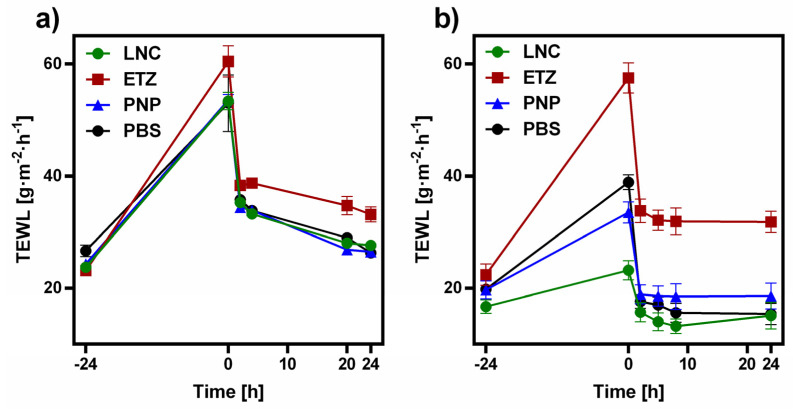
Development of trans-epidermal water loss values of untreated and formulation-treated (**a**) porcine skin, (**b**) human skin. The formulation was applied on the skin between −24 and 0 h of the experiment. *n* ≥ 3.

**Figure 9 pharmaceutics-15-00513-f009:**
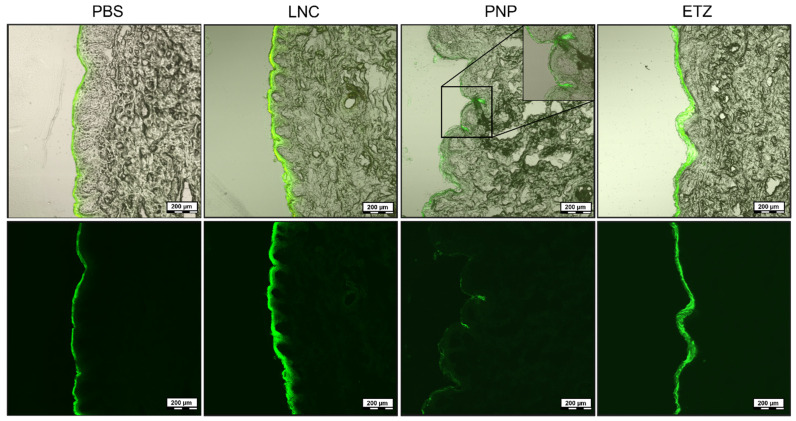
Confocal microscopy of skin treated for 24 h by a suspension of fluorescein sodium suspension in PBS, FNa-labelled lipid nanocapsules (LNC), FNa-labelled PLGA nanoparticles (PNP), and NBD-PC labelled ethosomes (ETZ). The scale bar represents 200 µm.

**Table 1 pharmaceutics-15-00513-t001:** The summary of advantages and disadvantages of nanosystems presented in this work.

Nanosystem	Advantages	Disadvantages
Lipid Nanocapsules (LNC)	Biocompatible composition	Higher surfactant content
	Skin occlusion effect	Difficult surface functionalization
	Drug delivery to deep skin layers	Complicated composition
	Great colloidal stability	
Polymeric Nanoparticles (PNP)	Particle size selection	Diluted formulations
	Natural polymers	Short colloidal stability
	Increased follicular targeting	Polymer biocompatibility issues
		Lower encapsulation rates
Ethosomes (ETZ)	Flexible nanovectors	Often high alcohol content
	Encapsulation of both hydrophilic and lipophilic drugs	Skin dryness
	Simple preparation process	Particle stability issues

**Table 2 pharmaceutics-15-00513-t002:** The composition of lipid nanocapsules prepared in this work for topical delivery of hydrocortisone (HC) and hydrocortisone-butyrate (HCB).

Compound	Amount [wt%]
Isopropyl myristate	15.00
Kolliphor HS 15	9.4
Phospholipon 90G	0.60
NaCl	1.70
API	0.10
PBS (pH 7.4)	28.8
PBS (freezer)	44.4

**Table 3 pharmaceutics-15-00513-t003:** The composition of polymeric nanoparticles and ethosomes prepared for topical delivery of hydrocortisone (HC) and hydrocortisone-17-butyrate (HCB).

Polymeric nanoparticles (PNP)	PLGA [wt%]	PVA solution (*v*/*v*)	HC [wt%]	HCB [wt%]
	3.33	0.5%	0.1	
	7.5	2%		0.05
Ethosomes (ETZ)	Phospholipon 90G [wt%]	Ethanol:PBS solution (*v*/*v*)	HC [wt%]	HCB [wt%]
	3	50%	0.1	0.1

**Table 4 pharmaceutics-15-00513-t004:** Particle size (hydrodynamic diameter), polydispersity index (PDI), and encapsulation efficiency (EE) of nanosystems containing hydrocortisone (HC) or hydrocortisone-17-butyrate (HCB) used in in vitro and ex vivo studies.

	HC			HCB		
	Size [nm]	PDI	EE%	Size [nm]	PDI	EE%
LNC	76 ± 7	0.09 ± 0.01	80 ± 2	82 ± 4	0.05 ± 0.01	86 ± 1
PNP	186 ± 4	0.04 ± 0.02	27 ± 2	211 ± 6	0.03 ± 0.01	78 ± 1
ETZ	264 ± 9	0.55 ± 0.08	n/a	153 ± 9	0.10 ± 0.02	n/a

## Data Availability

Data are contained within the article.

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
