# Peer review of "The Contest of Nanoparticles: Searching for the Most Effective Topical Delivery of Corticosteroids"

_pharmaceutics, 2023, doi:10.3390/pharmaceutics15020513_

Round 1

Reviewer 1 Report

The manuscript ID pharmaceutics-2101792 mainly studies the formulation of hydrocortisone and hydrocortisone-17-butyrate into lipid nanocapsules, polymeric nanoparticles and ethosomes. The characterization was carried out by dynamic light scattering, FTIR, confocal microscopy and transmission electron microscopy. Interesting results are presented but several issues should be addressed. Please see below a list of comments to the authors:

  1. Advantages and disadvantages of the different particles studied could be summarized in a table.
  2. A graphical abstract included in the body text could improve the justification and presentation of the aim of the research.
  3. Is there potential evolution of the physical properties exhibited by the samples over the main findings? Please argue.
  4. Please comment about reproducibility of the samples studied and the effect of the nanoformulations on the skin barrier.
  5. How is considered that agglomeration of the nanoparticles can influence the main results?
  6. Perspectives of this research should be mentioned to contemplate future potential applications confronted with the main findings. The authors are invited to see for instance  https://doi.org/10.1016/j.ijleo.2021.167738 and https://doi.org/10.1016/j.biopha.2022.112633
  7. The presentation of some references in collective citation form should be split in individual form in order to better justify the selection of the citations presented.
  8. The keywords should be separated by semicolon.
  9. The equations must be numbered.
  10. The fonts in confocal microscopy should be enlarged.

Author Response

Thank you very much for your valuable comments. Please, find our response in the attached file.

Reviewer 2 Report

Aneta Kalvodová and co-authors have done very interesting study wherein they have developed lipid nanoparticles, polymeric nanoparticles and ethosomes for topical delivery of two corticosteroids. For all these nano formulations, the route of penetration has been investigated. The studies performed are well organised and manuscript has been written nicely. The manuscript can be accepted for the publication.

However, it could have been more realistic approach if one particular polymer, lipid and their combinational system as in one carrier were investigated.   

Good luck!

Author Response

Thank you very much for your positive feedback. We agree that it is useful to compare systems of similar materials and combined nanoparticulate formulations. We plan to bear in mind this aspect in our following research.  

Reviewer 3 Report

The manuscript reports about a comparative study on three type of nanovectors (LNC, PNP, and ETZ). The systems are tested ex vivo on porcine and human skin for the dermal and trans-dermal delivery, using two drugs, HC  and HCB. The effects on the skin samples and theire reversibility were also characterized. The research is interesting, the experimental work is well-planned and extensive, the presentation is quite clear. Nevertheless, many comments need to be addressed before publication.

Major comments

1. The literature cited in the Introduction is not up-to-date, please provide more recent references and, if necessary, adjust the Introduction in accordance with recent advances.

2. In the introduction, authors should briefly review the state-of-the-art of use of LNC, PNP, and ETZ as trans-dermal delivery systems.

3. The word "size" is not accurate: since DLS measurements were performed to characterize the samples, please specify if hydrodynamic radius or hydrodynamic diameter values are reported in the manuscript. Also, specify the meaning of "Z-average" reported in line 174.

4. In section 2.4.3, report a sketch of the experimental setup to help comprehension.

5. In the description of TEWL experiments, more technical details should be provided; in particular, specify which quantity was measured (lines 291-292, the steady-state of what did authors note?) and which instruments was used.

6. Lines 390-392 "more lipophilic HCB is limited to hydrophobic interactions with the polymer. Based on these reasons, higher amount of PLGA and PVA was used for encapsulation of HCB into the PNP." It is not clear to me why it was necessary to increase also the amount of PVA that is hydrophilic, could you elucidate more this aspect?

7. Figure 1. It is not clear the reason for reporting measurements at two temperatures in the case of LNC, since they are not discussed in the manuscript. If not useful, remove the measurements at 4°C. Instead, for application purposes, it would be more interesting to use as second temperature 37°C instead of 25°C.

8. Lines 408-410 "The stability study of ethosomes was carried out by storing the samples in refrigerator (4 °C) or at laboratory temperature (25 °C) for 60 days and observing the development of particle size and PDI. The stability study results are shown in Figure 1". In Figure 1 the results at the two temperatures are not reported, please add them.

9. Lines 413-415 "Our prepared ethosomal formulation showed a negative zeta potential as expected due to the presence of ethanol, which causes negative charge, thereby the vesicles are prevented from aggregation and drug leak during storage". The values of PZ reported in Figure 1 (between -4 and -2 mV) are not high enough to support this statement: it is in fact possible to observe aggregation of nanoparticles even at extremely higher potentials, depending on the different interactions acting between colloids (see, for example doi.org/10.1016/j.jcis.2020.07.006).  The colloidal stability of the nanoparticles is instead ascertained by the hydrodynamic radius (or diameter?) measurements. Modify this sentence to be more precise.

10. It would be useful, for the scope of this research, to evaluate the rate of drug release of the different nanovectors analyzed (see, for example 10.1016/j.colsurfb.2019.05.006).

11. Lines 439-442 "Establishing the EE in a liposomal formulation is often problematic and may produce uncertainties in the methodology and computations. Lipophilic API should tend to attach to or into the phospholipid bilayer helping it with the solubilization and in consequence with higher EE." This period is not clear, what is the problem if the API attaches to the bilayer? Even if it is not inside the capsule, it will be carried by it through the skin. If not, please elaborate on this point to be more clear. In any case, the values of EE and DL should be reported in the manuscript.

12. FTIR results: provide a table with peaks assignment.

13. Based on the absence of the peaks assigned to API, authors claim in section 3.1.3 about "drug incorporation into the nanoparticle body". How can they exclude, based on these results, the total absence of the drug within the sample? Also, it is hard to follow the argument of the next sentence: "As PNP and ETZ showed much lower EE for HCB, this fact was shown in their individual spectra. In both cases HCB-PNP and HCB-ETZ contained bands from both, the drug and formulation components.": do authors mean that a lower amount of molecules determines higher peaks in the FTIR spectra? Please, revise the discussion of FTIR results.

14. Size measurements based on TEM images must be based on large statistics instead of selecting only the nanoparticles with sizes similar to those measured by DLS.

15. According to the discussions of results reported in Figure 6, it is not clear the meaning of the errorbars: the authors state that "LNC and PNP also had an effect on the skin barrier" even if the values reported in the plot appear compatible with 1; also they state that "the most significant effect on the skin was caused by the application of ETZ" while the value appears compatible with 1. Please, clarify how did you evaluate errorbars and revise the discussion accordingly.

n-1. Supplementary information was not available for reviewing.

n. Since human skin was used in experiments, please double-check compliance with the MDPI ethical guidelines.

Minor comments

- Report the detailed procedure used to evaluate the positions of peaks in FTIR spectra.

- Representation of data should be uniformed in section 3.1.1: move the DLS and ZP results from Table 3 to Figure 1.

- The modifications observed in the size of ETZ formulations after the first week should be discussed more deeply, in particular in the case of HCB. Which are the possible causes of such increase/decrease?

- To make Figure 2 more comprehensible: (i) increase the font size and lines thickness; (ii) label the spectra with the name of the samples as used in the text: LNC instead of IPM, PNP instead of PLGA, and ETZ instead of PL; (iii) specify in the caption all the colors used for spectra representation; (iv) specify in the caption the difference between "+" and "-" used in the sample labels.

- Figure 4 and Figure 5: (i) increase the font size; (ii) report the name of the quantity plotted in panels e and f, not only units; (iii) for better comparison, use the same y scale in the corresponding plots of the two columns.

- Figure 6: place errorbars over the bars of the histogram to make visible also their lower part.

- Figure 7: fist derivative plots can be removed, since they are not discussed in the text.

See also the other minor comments in the attached file.

Author Response

(The authors gave the same response as above.)

Round 2

Reviewer 1 Report

The authors have provided a reviewed version that still presents some issues that should be addressed.

Important points should be deeply studied: the advantages and disadvantages of the different particles summarized in a table in the claimed contest of the title of the article is missing, the evolution of the physical properties should be better described for reporting the main findings, and the perspectives of this research should be supported by a critical confrontation assisted by publications.

Reviewer 3 Report

1. The concern about the word “size” was not addressed satisfactorily, as (i) the word is ambiguous as it is not clear if the measured quantity is the hydrodynamic radius or the hydrodynamic diameter, and (ii) the expression “Z-average” identifies the procedure used by the software to derive the hydrodynamic radius (or diameter?) from the measured correlograms and not to a physical quantity, it is therefore incorrect to use it throughout the manuscript. Therefore, authors should authors should keep the expression “Z-average” in the methods section, when they describe the procedure used for data analysis, and instead use the name of the specific quantity reported (“hydrodynamic radius” or “hydrodynamic diameter”) throughout the the other parts of the manuscript.

2. Authors state, in their reply, that the scheme of the experimental setup used in permeation experiments ex vivo is reported in the graphical abstract, but no graphical abstract was provided: they should show it.

3. Regarding the evaluation of the positions of peaks in FTIR spectra, it would be appropriate to specify the fitting function selected in the procedure.

4. In Figure 1, add to the legend the black dots representing PDI values.

5. In lines 540-541, the sentence “where no significant change was observed in the value of Z-average, PDI and ZP” is not accurate, as the reported quantities show significant variations, as least after 30 days: please, amend it.

6. Figure 2 was not modified satisfactorily: (i) text size was not increased sufficiently, (iii) it is appropriate to use the names of the components for the physical mixtures, but the names of the vectors should be used for the corresponding spectra.

7. In Figure 4 and Figure 5, the cuts in the y axes should be avoided (or uniformed) for proper comparison between left and right columns.

8. The Section “Supplementary Materials” (lines 1045-1047), should be double checked to make it consistent with the Figures and the Tables present in the submitted file.

Round 3

Reviewer 1 Report

The authors have improved the presentation of their work. I can recommend this work for publication in present form.